# Mitochondrial Dysfunction in Skeletal Muscle of Rotenone-Induced Rat Model of Parkinson’s Disease: *SC-*Nanophytosomes as Therapeutic Approach

**DOI:** 10.3390/ijms242316787

**Published:** 2023-11-27

**Authors:** Daniela Mendes, Francisco Peixoto, Maria Manuel Oliveira, Paula Branquinho Andrade, Romeu António Videira

**Affiliations:** 1REQUIMTE/LAQV, Laboratory of Pharmacognosy, Department of Chemistry, Faculty of Pharmacy, University of Porto, Rua de Jorge Viterbo Ferreira, nº 228, 4050-313 Porto, Portugal; mendesdaniela04@hotmail.com (D.M.); pandrade@ff.up.pt (P.B.A.); 2Chemistry Center-Vila Real (CQ-VR), Biological and Environment Department, School of Life and Environmental Sciences, University of Trás-os-Montes e Alto Douro, UTAD, P.O. Box 1013, 5001-801 Vila Real, Portugal; fpeixoto@utad.pt; 3Chemistry Center-Vila Real (CQ-VR), Chemistry Department, School of Life and Environmental Sciences, University of Trás-os-Montes e Alto Douro, UTAD, 5001-801 Vila Real, Portugal; mmso@utad.pt

**Keywords:** elderberry anthocyanins, algae polar lipids, mitochondrial dysfunction, skeletal muscle, nanomedicine, Parkinson’s disease

## Abstract

The development of new therapeutic options for Parkinson’s disease (PD) requires formulations able to mitigate both brain degeneration and motor dysfunctions. *SC-*Nanophytosomes, an oral mitochondria-targeted formulation developed with *Codium tomentosum* membrane polar lipids and elderberry anthocyanin-enriched extract, promote significant brain benefits on a rotenone-induced rat model of PD. In the present work, the effects of *SC-*Nanophytosome treatment on the skeletal muscle tissues are disclosed. It is unveiled that the rotenone-induced PD rat model exhibits motor disabilities and skeletal muscle tissues with deficient activity of mitochondrial complexes I and II along with small changes in antioxidant enzyme activity and skeletal muscle lipidome. *SC-*Nanophytosome treatment mitigates the impairment of complexes I and II activity, improving the mitochondrial respiratory chain performance at levels that surpass the control. Therefore, *SC*-Nanophytosome competence to overcome the PD-related motor disabilities should be also associated with its positive outcomes on skeletal muscle mitochondria. Providing a cellular environment with more reduced redox potential, *SC*-Nanophytosome treatment improves the skeletal muscle tissue’s ability to deal with oxidative stress stimuli. The PD-related small changes on skeletal muscle lipidome were also counteracted by *SC-*Nanophytosome treatment. Thus, the present results reinforces the concept of *SC-*Nanophytosomes as a mitochondria-targeted therapy to address the neurodegeneration challenge.

## 1. Introduction

Parkinson’s disease (PD) is a degenerative brain disease with a clinical pattern of progressive motor impairment characterized by bradykinesia, tremor, rigidity, and imbalance. During disease progression, PD patients also exhibit many non-motor symptoms including sleep disturbances, depression, cognitive dysfunction, and dementia [1,2]. Clinical symptoms are usually attributed to a gradual loss of functional connectivity between basal ganglia–cortex–cerebellum, brain regions dominated by dopamine-producing neurons that play a key role in turning movement thought into action [3]. Thus, the approved therapeutics were rationalized to improve dopamine-dependent connectivity. Despite their symptomatic benefits in the first stages of treatment, the dopamine precursors (e.g., carbidopa–levodopa) individually or in combination with the inhibitors of monoamine oxidase-B inhibitors (e.g., selegiline, rasagiline, and safinamide) are unable to stop or to reverse the degenerative process, becoming the problem of PD as one of the greatest challenges of our time [4,5].

Considering the limitations of approved therapeutics, PD research has been directed towards therapeutic approaches targeting the disease’s sub-cellular pathological hallmarks, such as aggregation, or clumping, of the protein α-synuclein, loss of cell redox balance and mitochondrial dysfunction [6,7]. Mitochondria-targeted therapies, mainly designed to modulate the functioning of the mitochondrial respiratory chain, are very promising in combating PD. Firstly, mitochondrial dysfunctions, characterized by impaired activity of the mitochondrial respiratory chain (mainly complex I, NADH:ubiquinone oxidoreductase), are detected from the earliest stages of the disease [1]. Secondly, mitochondrial dysfunctions are strictly associated with oxidative stress and misfolded protein accumulation, including α-synuclein in cells that express this protein, as well as many other metabolic-related symptoms. Thirdly, xenobiotics that work as specific mitochondrial complex I inhibitors (*e.g.,* rotenone, and MPTP 1-methyl-4-phenyl-1,2,3,6-tetrahydropyridine) promote a PD-like neurodegenerative process [8,9].

Working within this framework, we developed *SC-*Nanophytosomes, a mitochondria-targeted nanoplatform built with algae membrane polar lipids from *Codium tomentosum* and elderberry anthocyanin-enriched extract (EAE-extract) from *Sambucus nigra* [10]. *C. tomentosum* membrane polar lipids are a particular set of lipids dominated by anionic phospholipids (mainly phosphatidylglycerols, PG) containing a fatty acid profile with high levels of polyunsaturated fatty acids (preferentially n−3 PUFA) [10,11]. This set of lipids supports the development of nanocarriers with soft architecture and competence to accommodate the elderberry anthocyanins in their flavylium cationic form [10]. Moreover, this nanocarrier can also work as a source of lipids with physiological relevance in the context of mitochondrial dysfunction and neurodegeneration. PG is a cardiolipin precursor, the lipid fingerprint mitochondria required to ensure the mitochondrial redox chain functionality [12,13], and the high n−3/n−6 PUFA ratio is associated with anti-inflammatory outcomes [14].

On the other hand, the cationic form of elderberry anthocyanins is a mitochondriotropic compound that exhibits redox reversibility and the ability to oxidize nicotinamide adenine dinucleotide (NADH) and deliver the electrons for mitochondrial complex III (cytochrome *c* reductase). Thus, elderberry anthocyanins work as a membrane electron carrier, bypassing the abnormalities associated with impaired complex I activity [15], the main mitochondria-related pathological hallmarks of PD. In fact, *SC-*Nanophytosomes not only preserve the bioactivity of its building blocks, but they also enhance the positive outcomes of the building blocks in the context of neurodegeneration, as revealed by in vitro and in vivo studies. Using cellular models, we have shown that *SC-*Nanophytosomes protect SH-SY5Y cells from the toxicity promoted by rotenone or glutamate (compounds with a key role in neurodegenerative diseases), improving mitochondrial functionality and reducing the cellular generation of reactive oxygen species (ROS) [10]. Using a rotenone-induced rat model of PD, it was revealed that *SC-*Nanophytosomes, delivered by oral route, reduce the brain levels of α-synuclein, mitigate the brain mitochondrial dysfunction, improve the cell redox state and the functionality of the mitochondrial lipidome [16].

Skeletal muscle atrophy plays an important role in the progressive severity of the motor symptoms that characterize PD [17]. Although the PD symptoms are mainly associated with the degenerative brain process, the skeletal muscle of PD patients also exhibited several brain pathological hallmarks of the disease, including mitochondrial dysfunction related to deficient activity of the mitochondrial redox chain and the accumulation of misfolded α-synuclein [18,19]. Additionally, exercise programs that improve mitochondrial plasticity, biogenesis, and respiration promote benefits in patients with PD [20], highlighting the role of the skeletal muscle–brain axis on health and diseases of the brain [21]. Thus, the present work aims to characterize the effects of the oral administration of *SC-*Nanophytosomes on the skeletal muscle tissues of a rotenone-induced rat model of PD to reinforce the concept of mitochondria-targeted therapy for PD, developed considering brain benefits of the formulation on this set of animals [16].

## 2. Results and Discussion

### 2.1. In Vivo Assays—Benefits of SC-Nanophytosomes Formulation on Associated Motor Symptoms of the Rotenone-Induced Rat Model of PD

*SC-*Nanophytosomes are nanosized vesicles with a highly negative surface charge that exhibit stability when stored for 14 days at 4 °C [10] and preserve their properties under external pH changes that mimic the gastrointestinal tract conditions [16]. Thus, for the treatment of animals with a PD-like pathology, three independent *SC-*Nanophytosomes formulations were prepared, one per week, built with 600 μM of algae membrane polar lipids and 0.5 mg/mL of EAE-extract. These formulations were characterized in terms of anthocyanin entrapment efficiency, vesicle size, and surface charge. After 1 h of dialysis against a large volume of buffer, the *SC-*Nanophytosomes exhibit an anthocyanin entrapment efficiency of 81.39 ± 0.43%, a size of 96.84 ± 14.28 nm and a surface charge, assessed by Zeta potential, of −45.66 ± 9.14 mV.

A rotenone-induced rat model of PD exhibits many motor and non-motor pathological features of human disease, including resting tremor, postural instability, and brain accumulation of α-synuclein and mitochondrial dysfunction [16]. Therefore, the rotenone-induced rat model of PD exhibits the main phenomenology of PD patients [22]. Thus, the animals of the three experimental groups were submitted to a beam walking test at the beginning and once a week during the three weeks of treatment with *SC-*Nanophytosomes to assess PD-related motor symptoms. Videos recorded during the beam walking test show that ROT group animals exhibit tremors, postural instability, balance disturbance, and a visible dragging of rear legs, symptoms which were not detected in the animals of the CTRL group (Video S1). The benefits of *SC-*Nanophytosome treatment on PD-related motor symptoms are well evidenced in the videos recorded during the beam walking test by the absence of tremors and by reducing the imbalance (Video S1). Therefore, the present video confirms that the rotenone-induced PD animal model exhibits significant motor impairment detected and *SC-*Nanophytosomes treatment promote benefits in motor coordination and balance, as previously described considering the performance of this set of animals in beam walking test [16].

### 2.2. Ex Vivo Assays in Skeletal Muscle Tissues-Effects on Mitochondrial Redox Chain Complex Functionality, Cell Redox State and Fatty Acid Profile

Despite the PD motor impairments, major concerns in disease progression have been attributed to neuronal degeneration issues, and not muscle contractile capacity. The pathological brain hallmarks of PD, including mitochondrial dysfunction and oxidative stress, are also detected in human patients’ skeletal muscle tissues and animal models for PD [18,23]. Mitochondrial dysfunctions and oxidative stress are interlinked factors that compromise not only the bioenergetic metabolism but also the lipid metabolism, which can be detected by changes in membrane lipid profile [24]. Thus, the metabolic changes in skeletal muscle tissues triggered by oxidative stress and mitochondrial dysfunction should also contribute to motor impairment with a pattern of progressive severity exhibited by PD patients. Considering this framework, the mitigation of PD-related skeletal muscle pathological hallmarks should be considered in assessing the effectiveness of any therapeutic intervention. Thus, the therapeutic potential of *SC-*Nanophytosomes as mitochondria-targeted therapy for PD was assessed in skeletal muscle tissues, considering effects on mitochondria functionality, cell redox state and fatty acid profile.

#### 2.2.1. Effects on the Activity of Mitochondrial Redox Chain Complexes

Mitochondrial respiratory failure in the skeletal muscle of patients with PD has been reported [18,19], emphasizing their role in motor symptoms. Thus, the benefits of the oral administration of *SC-*Nanophytosomes on the skeletal muscle mitochondria were assessed considering the effects on the activity of the individual respiratory complexes I, II (succinate dehydrogenase), and IV (cytochrome c oxidase) in the mitochondria-rich fraction (Figure 1). The activity of these mitochondrial complexes was normalized by the citrate synthase (CS) activity to discard putative differences in the mitochondria content of the samples. Skeletal muscle mitochondria obtained from three experimental groups exhibit similar values of CS activity, namely 134.95 ± 16.05, 137.19 ± 16.45, and 106.30 ± 12.78 for CTRL, ROT, and ROT+*SC-*Nanophyt groups, respectively.

The results in Figure 1 show that the mitochondrial complex I and II activities decreased in the ROT group compared to the CTRL group. The results showed no difference between the CTRL and ROT groups regarding mitochondrial complex IV activity. The impairment of both complexes I and II in the skeletal muscle of the ROT group animals suggests that the neurodegenerative process underlying PD is associated with a mitochondrial dysfunction extendable beyond the brain, which is detected in peripheral organs like skeletal muscle as reported here for a rotenone-induced PD rat model (Figure 1), and previously for other models [25,26]. In fact, the brain and skeletal muscle are among the most metabolically active tissues. Consequently, their functional performance is particularly vulnerable to mitochondrial impairment [27], and the bidirectional connectivity between brain and skeletal muscle is also well-documented [21]. Therefore, skeletal muscle mitochondrial dysfunction detected in the ROT group can emerge either by the direct effects of rotenone on skeletal muscle or by its degenerative brain effects.

Regarding the effects of *SC-*Nanophytosomes treatment (Figure 1), the activities of complexes I, II, and IV in the ROT+*SC-*Nanophyt group are significantly increased when compared with the ROT group, indicating that *SC-*Nanophytosomes have the competence to restore the impairment of mitochondrial respiratory complex activities promoted by rotenone. Moreover, the mitochondria of skeletal muscle tissues of the ROT+*SC-*Nanophyt group exhibit activities of complexes II and IV significantly higher than mitochondria of the CTRL group, suggesting that the treatment could improve the mitochondrial respiration performance at a level that surpasses the control animals. These effects align with the mitochondrial bioactivity of elderberry anthocyanins revealed in isolated mitochondria, cells, and the brain of this set of animals [10,15,16].

#### 2.2.2. Effects on the Cell Redox State

In general, oxidative stress results from an excessive generation of ROS and/or from deficient activity of the antioxidant enzymes. Oxidative stress biomarkers mirror the unbalance of cell redox status, mainly characterized by a decrease in the reduced glutathione (GSH)/oxidized glutathione (GSSG) ratio. Therefore, the capacity of *SC-*Nanophytosome treatment to modulate the redox status of skeletal muscle tissues of the rotenone-induced rat model of PD was evaluated in mitochondria-free cytosolic fractions, measuring the levels of GSH and GSSG (Figure 2) and the activity of antioxidant enzymes, namely superoxide dismutase (SOD), catalase (CAT), glutathione peroxidase (GPx), and glutathione reductase (GR) (Figure 3). The integrated assessment of these enzymatic and nonenzymatic antioxidant parameters allows an overview of the response of skeletal muscle tissues to oxidative stress stimuli, such as rotenone used to induce PD-like pathology. While the GSH/GSSG ratio is used as an indicator of the redox state of cells, antioxidant enzymes are key players in the management of the ROS generated in the cells, thereby the activity of antioxidant enzymes is interlinked with intracellular GSH/GSSG ratio [28,29,30].

Figure 2 shows that the cytosolic levels of GSH are similar in the three experimental groups and that GSSG levels in the ROT+*SC-*Nanophyt group are similar to the CTRL group but significantly lower than in the ROT group. The GSSG levels in the ROT group slightly increased compared to the CTRL group, but the differences did not reach statistical significance. Consequently, the GSH/GSSG ratio in the ROT+*SC-*Nanophyt group is significantly higher than in the ROT group (*p* < 0.01), but it is not statistically different from the CTRL group. These results suggest that the PD-like pathology induced by rotenone has a minor impact on the redox status of skeletal muscle tissues and that treatment with *SC-*Nanophytosomes promotes a cytosolic environment with more reduced redox potential (high GSH/GSSG ratio), enhancing the skeletal muscle cell competence to lead with oxidative stress stimuli.

Regarding the activity of SOD, CAT, GR, and GPx enzymes in the cytosolic fraction of the skeletal muscle tissues, data from Figure 3 show that the rotenone-induced rat model of PD only affects the activity of CAT, promoting a significant increase in its activity when compared to the CTRL group. The increased CAT activity in the ROT group suggests that the skeletal muscle cells reply to a cytosolic increase in hydrogen peroxide levels, enhancing the activity of this enzyme. Additionally, the treatment with *SC-*Nanophytosomes in rats with PD-like pathology strongly impacts the activity of all evaluated antioxidant enzymes. Thus, in the ROT+*SC-*Nanophyt group, a significant decrease in SOD, CAT, GR, and GPx enzyme activity compared with the CTRL group or the ROT group is detected. Previously, it was shown that the *SC-*Nanophytosomes have positive brain impacts on the rotenone-induced PD rat model by modulating the activity of antioxidant enzymes in brain cells [16].

From the data in Figure 2 and Figure 3, we can conclude that the treatment with *SC-*Nanophytosomes has the competence to increase the GSH/GSSG ratio simultaneously with a decrease in the antioxidant enzyme activities. Therefore, the *SC-*Nanophytosome treatment mitigates the effects of the rotenone-induced PD-like pathology, mainly detected by CAT activity, providing a cellular environment with a more reduced redox potential, which has high relevance in the context of degenerative diseases and the ageing process [31].

#### 2.2.3. Effects on the Fatty Acid Profile of the Skeletal-Muscle Tissue

The next step is to investigate if *SC-*Nanophytosomes have therapeutic benefits by modulating the skeletal muscle lipidome. Lipid studies have high relevance in the PD context since mitochondria functionality, synaptic signaling, and endosome–lysosome system functioning, all affected by the degenerative process, rely heavily on the membrane lipid composition of the cells and tissues affected by the disease [32,33,34,35]. Thus, the characterization of the fatty acid profile of skeletal muscle tissues of the three experimental groups allows us to identify lipid changes connected with the rotenone-induced PD-like pathology and to reveal the putative effects of *SC-*Nanophytosome treatment. Figure 4 and Figure 5 show the fatty acid profile of the total membrane lipid extracted from the skeletal muscle tissues of the CTRL, ROT, and ROT+*SC-*Nanophyt groups.

As shown in Figure 4, the skeletal muscle fatty acid profile of the CTRL group comprises twenty-five different species, of which seven are saturated fatty acids (SFA), five monounsaturated fatty acids (MUFA), and thirteen PUFA. Palmitic acid (C16:0) and stearic acid (C18:0) species dominate the SFA. In contrast, oleic acid (C18:1) is the most abundant MUFA, followed by palmitoleic acid (C16:1). Linoleic acid (C18:2n−6) is the dominant species of PUFA and the most abundant molecular species in the n-6 series followed by the arachidonic acid (AA, C20:4n−6), while docosahexaenoic acid (DHA, C22:6n−3) is the most abundant in the n−3 series. According to these results, Abbott and collaborators [36] showed that DHA accounted for most of the n−3 series in the rats’ muscle lipidome. At the same time, C18:2n−6 and AA provided the highest relative abundance in the total n−6 series. Regarding the fatty acid profile of the ROT group, a significant increase in C18:0, DHA, and AA content is detected, and a significant decrease in the levels of C16:0, C18:1, and C18:2n−6 when compared with the CTRL group. The treatment with *SC-*Nanophytosomes, ROT+*SC-*Nanophyt group, promotes a reversion of the effects promoted by rotenone, bringing the levels of the affected species to values close to the control. Thus, it is detected in the ROT+ *SC-*Nanophyt group, with a significant increase in C16:0, C18:1, and C18:2n−6 content, and a significant decrease in the levels of C18:0 and AA, when compared with the ROT group.

Figure 5 shows the levels of SFA, MUFA, n−3, and n−6 PUFA as well as the lipid-related functional parameters represented by unsaturation index and n−3/n−6 PUFA and DHA/AA ratios. Thus, the ROT group has a fatty acid profile with higher PUFA and SFA richness and lower levels of MUFA when compared with the CTRL group. However, the differences only reach statistical significance for MUFA. No significant differences between CTRL and ROT groups are detected in the n−3/n−6 PUFA and DHA/AA ratios and unsaturation index parameters. Despite the *SC-*Nanophytosome treatment promoting quantitative changes in fatty acid profile, the impact on the parameters mentioned above is minor. The differences did not reach statistical significance when compared with the CTRL or ROT groups. However, the DHA/AA and n−3/n−6 PUFA ratios in the ROT+*SC-*Nanophyt group are closer to the CTRL group, suggesting that the treatment counteracts the small changes promoted by rotenone on skeletal muscle lipidome.

## 3. Materials and Methods

### 3.1. SC-Nanophytosome Preparation and Characterization

*SC-*Nanophytosomes were prepared in buffer solution (50 mM KCl, 10 mM HEPES, 2 mM citric acid, pH 6.4) with EAE-extract and *C. tomentosum* polar membrane lipids at 0.5 mg/mL and 600 μM in phospholipids [10]. *C. tomentosum* polar membrane lipids extract contains a set of lipids composed of 26% diacylglycerol lipids without phosphorus and 74% of phospholipids, of which 20,0% are phosphatidylcholine, 13.8% phosphatidylinositol, 5.7% phosphatidylserine, 22.5% PG and 6.9% phosphatidic acid. This set of lipids has a fatty acid profile with 34% of PUFA with an n-3/n-6 PUFA ratio of 1.5 [10]. EAE-extract contains four anthocyanins (cyanidin-3-*O*-glucoside, cyanidin-3-*O*-sambubioside, cyanidin-3,5-*O*-di-glucoside, and cyanidin-3-*O*-sambubioside-5-*O*-glucoside), which together account for 86% of the total phenolic content, quercetin-3-*O*-glucoside, and small amounts of a *p*-hydroxybenzoic acid derivative [10].

*SC-*Nanophytosomes were characterized by anthocyanin entrapment efficiency, size, and surface charge. Anthocyanin entrapment efficiency was assessed by spectroscopy, recording the UV-Visible spectra of the *SC-*Nanophytosome formulations before and after remotion of the non-encapsulated polyphenols by dialysis through a cellulose membrane (cut-off of 14 kDa, Sigma-Aldrich, St. Louis, MO, USA) against a large volume of aqueous buffer solution (50 mM NaCl, 10 mM HEPES, 2 mM citric acid, pH 6.4), as previously described [10]. The size of the dialyzed *SC-*Nanophytosomes was evaluated by dynamic light scattering (DLS), while the surface charge, measured by Zeta potential, was evaluated by electrophoretic light scattering (ELS) [10].

### 3.2. Rotenone-Induced Rat Model of PD, Treatment with SC-Nanophytosomes and the Beam Walking Test

The animal experimental plan was compliant with Portuguese (Decreto-Lei 113/2013) and European (EU Directives 2010/63/EU) guidelines for Animal Care and was licensed by the Portuguese Veterinarian and Food Department (approval nº 0421/2020) and approved by the Ethics Committee of UTAD. The detailed description of conditions and experimental plan were described in our previous work [16], where it was reported the brain effects of treatment with *SC-*Nanophytosomes on the rotenone-induced rat model of PD, using this set of animals. The present work is focused on skeletal muscle effects. The study was carried out with three experimental groups with five animals each: control (CTRL), rotenone (ROT), and rotenone treated during three weeks with *SC-*Nanophytosomes via the oral route (ROT + *SC-*Nanophyt), under an experimental plan with two stages separated by one week. In the first step, rotenone (3.0 mg/kg of animal weight, i.p., three times a week for three weeks) was used to induce a PD-like pathology in 10 animals (ROT and ROT + *SC-*Nanophyt groups). Simultaneously, a similar procedure was used to inject the vehicle used to solubilize rotenone (liposomes prepared with egg yolk phospholipids) into the five animals of the CTRL group. In the second step, *SC-*Nanophytosome formulation, delivered by drinking water at a final concentration of 3 µM in phospholipids plus 2.5 mg/L in EAE-extract, was used to treat the five animals of the ROT + *SC-*Nanophyt group for three weeks. Considering the daily liquid intake and the animals’ body weight for the ROT + *SC-*Nanophyt group, the *SC-*Nanophytosome oral dose of 300 nmol of phospholipid and 0.25 mg of EAE-extract/kg/day was calculated [16]. The performance of all animals in the beam walking test, measured and video recorded once a week, was used to confirm the competence of the rotenone to induce a PD-Like pathology and to evaluate the therapeutic efficacy of *SC-*Nanophytosomes at level of the motor symptoms [16,37].

### 3.3. Ex Vivo Studies: Skeletal-Muscle Tissues Processing for Bioenergetic, Cell Redox State and Lipidomic Assays

After *SC-*Nanophytosome treatment, the animals were euthanized by cervical displacement followed by decapitation. Brain, heart, liver, kidneys (characterized in reference [16]), and samples of skeletal-muscle tissues were collected from the thighs of both hind limbs, immediately frozen in liquid nitrogen, and kept at −80 °C until to be used.

Skeletal-muscle samples, with a known mass, were cut into small pieces and homogenized in ice-cold buffer (sucrose 130 mM, KCl 50 mM, MgCl_2_ 5 mM, KH_2_PO_4_ 5 mM, and HEPES 5 mM, pH 7.4, supplemented with a protease inhibitors cocktail) (1:10 p/v) using a Glass-Teflon Potter Elvejhem. The homogenate was filtered through standard surgical gauze to remove the solid debris and the filtered homogenates were collected. One part of the filtered homogenate was quickly frozen in liquid nitrogen and stored at −80 °C until used for evaluation of the fatty acid profile. The remaining filtered homogenate was fractioned by differential centrifugation to obtain the mitochondria-free cytosolic and the mitochondria-rich fractions using the previously described procedures. The protein content of these sub-cellular fractions as well as the filtered homogenates was determined by the Biuret method [38,39].

### 3.4. Assessment of the Mitochondrial Respiratory Chain Complex Activities

The activity of mitochondrial redox chain complexes, complex I, II, IV, and CS was evaluated in the mitochondria-rich fraction obtained from the skeletal muscle using routine procedures [38,39]. Microplate spectrophotometric or spectrofluorimetric assays were used to assess the mitochondrial enzymes’ activity in the mitochondria-rich fraction (Synergy H1 Biotek). All assays were performed at 37 °C, in 250 µL of reaction medium supplemented with 10 µg of mitochondrial protein.

Complex I activity was evaluated following NADH oxidation, detected by the blue/cyan fluorescence decay at 450 nm under excitation light with 366nm. This activity was evaluated in phosphate buffer (KH_2_PO_4_ 25 mM, MgCl_2_ 5 mM, pH 7.5) supplemented with 1 mM potassium cyanide (KCN), 8 μM antimycin A, 0.775 mM decylubiquinone, 12 μM rotenone and 0.15 mM NADH. The fluorescence was measured in the absence and presence of rotenone (complex I inhibitor) and the results expressed as nmol NADH oxidized per minute per mg of protein.

Complex II was evaluated following the reduction of 2,6-dichlorophenolindophenol (DCPIP, color change from blue to colorless), promoted by reduced-ubiquinone, in the presence of the complex I inhibitor is only produced by electron delivered by the oxidation of the succinate by the enzyme. This activity was evaluated in phosphate buffer (KH_2_PO_4_ 25 mM, pH 7.5) supplemented with 0.25 mM DCPIP, 1 mM KCN, 8 μM antimycin A, 0.194 mM decylubiquinone, 12 μM rotenone, 10 mM oxaloacetate, and 2 mM succinate. The absorbance at 600 nm was measured in the absence and presence of oxaloacetate (complex II inhibitor) and the results expressed as nmol DCPIP reduced per minute per milligram of protein.

Complex IV activity was evaluated following the oxidation of reduced cytochrome *c* at 550 nm (color change from light red to dark red). Complex IV activity was evaluated in phosphate buffer (KH_2_PO_4_ 25 mM, pH 7.5) supplemented with 8 μM antimycin A, 12 μM rotenone, 1 mM KCN and 28 μM reduced Cyt *c*. The absorbance was measured in the absence and presence of KCN (complex IV inhibitor), and the results expressed as nmol oxidized Cyt *c* per minute per milligram of protein.

CS activity was determined by monitoring the color change associated with the reduction of 5,5-dithio-bis-2-nitrobenzoic acid (DTNB) by the CoASH, a product of the condensation reaction catalyzed by CS enzyme (acetyl-CoA plus oxaloacetate generate citrate and CoASH). CS activity was evaluated in buffer (Tris-HCl 20 mM, pH 8, Triton X-100 0.02%) supplemented with 0.4 mM DTNB, 2 mM oxaloacetate, 0.122 mM acetyl-CoA and 1.2 mM succinyl-CoA. The absorbance at 412 nm was measured in the absence and presence of succinyl-CoA (CS inhibitor) to determine the nmol TNB produced per minute per milligram of protein, which is stoichiometrically related to both CoASH and citrate generated by the enzyme. The activity of mitochondrial redox chain complexes was normalized by CS activity.

### 3.5. Determination of Enzymatic and Non-Enzymatic Antioxidant Defences in Skeletal Muscle Mitochondria-Free Cytosolic Fraction

The mitochondria-free cytosolic fraction obtained from skeletal muscle tissue was used to assess the activity of the SOD, CAT, GPx, and GR, as well as to determine the levels of GSH and GSSG [15,38,39]. Enzymatic assays are conducted at 37 °C using a volume of the mitochondria-free cytosolic fraction containing 10 µg of protein. The CAT activity was assessed polarographically using a Clark-type electrode (Hansatech, Norfolk, UK). At the same time, SOD, GR, GPx and the levels of GSH and GSSG were evaluated using a microplate reader (Synergy H1 Biotek, Winooski, VT, USA).

Cu/Zn-SOD activity was assessed in phosphate buffer (KH_2_PO_4_ 100 mM, EDTA 5 mM, pH 7.4) supplemented with 0.16 mM NBT, 0.5 U/mL xanthine oxidase, and 0.32 mM xanthine, measuring the purple formazan produced at 560 nm. In this methodology, the O_2_^•−^ generated by the xanthine/xanthine oxidase system can react with enzyme or with the yellow nitroblue tetrazolium chloride (NBT). Thus, when the O_2_^•−^ is generated by 1 enzyme’s catalytic activity (U) of xanthine oxidase, one catalytic unit of Cu/Zn-SOD is defined by the amount of biological sample required to decrease the slope of NBT reduction by 50%. The values were expressed as U per mg of protein.

CAT activity was determined following O_2_ production resulting from 1.76 mM H_2_O_2_ decomposition. CAT was evaluated in phosphate buffer (KH_2_PO_4_ 100 mM, EDTA 5 mM, pH 7.4) supplemented with H_2_O_2_. The CAT values were expressed in terms of nmol O_2_ produced per minute per milligram of protein.

GPx was evaluated through a coupled reaction with the GR enzyme. GPx reduces H_2_O_2_ to water and oxidizes GSH to GSSG, which is continuously reduced to GSH by an excess of GR with concomitant use of nicotinamide adenine dinucleotide phosphate (NADPH) in a stoichiometric amount, which can be assessed by fluorescence spectroscopy. GPx was evaluated in phosphate buffer (KH_2_PO_4_ 50 mM, EDTA 5 mM, pH 7.4) supplemented with 0.4 mM GSH, 4.66 U/mL GR, 0.2 mM NADPH, and 7.04 mM H_2_O_2_. The fluorescence was measured at 450 nm (setting excitation at 366 nm) in the absence and presence of GSH (substrate of GPx enzyme). The enzyme activity was expressed in nmol NADPH per minute per milligram of protein.

GR activity was evaluated following NADPH oxidation associated with the reduction of GSSG to GSH by fluorescence spectroscopy. GR activity was assessed in phosphate buffer (KH_2_PO_4_ 100 mM, EDTA 5 mM, pH 7.4) supplemented with 0.4 mM GSSG and 0.2 mM NADPH. The fluorescence was measured at 450 nm (setting excitation at 366 nm) in the absence and presence of GSSG (enzyme–substrate). The enzyme activity was expressed in nmol NADPH per minute per milligram of protein.

GSH and GSSG levels were measured by spectrofluorimetric assays, setting excitation at 339 nm and emission at 426 nm. GSH and GSSG were extracted from the mitochondria-free cytosolic fraction (0.5 mg/protein) by combining a lysing step (0.5% Triton X-100 in 5 mM HEPES) with an acidic treatment (H_3_PO_4_ 2.5%, m/V) under water-bath sonication to promote membrane rupture and protein denaturation, discarded by centrifugation (16,000× *g* for 30 min at 4 °C). GSH and GSSG levels were assessed in the collected supernatants after neutralization with NaOH solution. Supernatants were diluted in buffer (100 mM potassium phosphate, 5 mM EDTA, pH 8.0) supplemented with 266.67 μg o-phthalaldehyde (OPT). After incubation for 15 min, in the dark at room temperature, the mixture was used to determine the GSH levels. To determine GSSG content, supernatants were first incubated, in the dark, with 1.07 mM N-ethylmaleimide for 45 min. Then, were added 46.7 mM NaOH and phosphate buffer supplemented with 266.67 μg OPT followed by 15 min incubation at room temperature in the dark. Standard curves obtained with GSH and GSSG solutions with known concentrations were used to calculate the levels of GSH and GSSG in biological samples. The results were expressed as nmol GSH or GSSG per mg protein, and as GSH/GSSG ratio.

### 3.6. Analysis of Fatty Acid Methyl Esters from Skeletal Muscle Tissues

Total lipids were obtained from skeletal muscle tissue homogenates using a double-extraction procedure with a solvent combination of methanol/chloroform/water (2:1:0.8, *v*/*v*/*v*), and then the phospholipid content was determined according to the Bartlett and Lewis method, as previously described [40]. Fatty acid methyl esters (FAME) of total lipid extract (500 nmol in phospholipids) were obtained by acid-catalyzed transmethylation (5% (*v*/*v*) of HCL in methanol) in the presence of heptadecanoic acid (C17:0, 30 nmol), used as internal standard [41]. The fatty acid methyl esters solutions were analyzed by gas chromatography connected to mass spectrometry for species identification, or to a flame ionization detector for quantification, as described elsewhere [38].

### 3.7. Data and Statistical Analysis

Results are presented as mean values ± standard error (SEM) of at least three independent assays. One-way analysis of variance (ANOVA) together with the Bonferroni test was used to determine the level of significance between different groups. Statistical significance was attained at *p* < 0.05. Statistical analysis was performed and graphs in the figures were generated using the GraphPad Prism 8.0.1 software (San Diego, CA, USA).

## 4. Conclusions

Considering the present results with those obtained in the brain tissue of this same set of animals [16], we can assert that *SC-*Nanophytosomes exhibit structural and functional features to support a mitochondria-targeted therapy for PD covering the brain and peripheral tissues such as skeletal muscle. *SC-*Nanophytosomes exhibit properties that ensure stability in the main environment of the gastrointestinal tract and competence to preserve the anthocyanins in the form of flavylium cation, required to overcome PD-like mitochondrial dysfunction. Thus, *SC-*Nanophytosomes delivered via the oral route were rationalized as mitochondria-targeted therapy on a rotenone-induced rat model of PD. The treatment with *SC-*Nanophytosomes showed significant benefits in the motor disabilities promoted by rotenone with the improvement in motor coordination and balance. Although the skeletal muscle tissues of the rats with PD-like pathology only exhibit small changes in the cellular redox state, the higher activity of the CAT enzyme, compared with control animals, indicates that they are under oxidative stress stimuli. *SC-*Nanophytosome treatment leads to a redox potential of the skeletal muscle tissues towards more negative values (with a higher GSH/GSSG ratio), with a concomitant modulation of the activity of the main antioxidant enzymes. Thus, *SC-*Nanophytosome treatment provides an intracellular environment with a more reduced redox potential, improving the competence of the skeletal muscle tissues in counteracting oxidative stress damage. Furthermore, the *SC-*Nanophytosome treatment mitigates the impairment promoted by PD-like pathology on the mitochondrial redox chain complex activity and counteracts the changes in the fatty acid profile presented by this animal model of PD.

All these benefits can result from the competence of *SC-*Nanophytosomes to target algae lipids and elderberry anthocyanins to the mitochondria of skeletal muscle cells. The positive outcomes on the skeletal muscle mitochondrial respiratory chain and cell redox state can emerge from the elderberry anthocyanins, which have a positive charge and reversible redox properties. These properties allow those anthocyanins to work not only like common antioxidant molecules but also as a mitochondrial membrane electron carrier with the ability to oxidize NADH and deliver electrons for mitochondrial complex III, bypassing the mitochondrial dysfunction associated with impaired complex I activity [10]. However, the skeletal muscle positive outcomes exhibited by animals with rotenone-induced PD-like pathology treated with *SC-*Nanophytosomes can also emerge from an improved connection between brain and skeletal muscle resulting from the *SC-*Nanophytosomes brain effects previously described [16]. Several myokines and metabolites produced by skeletal muscle were identified as key players of the skeletal muscle–brain axis, which explains the experimental evidence that beneficial effects on skeletal muscle functionality positively impact brain disorders, including PD [21,42].

In conclusion, the present results reinforce the concept of *SC-*Nanophytosomes as a mitochondria-targeted therapy for PD, extending positive effects to the skeletal muscle tissues. Therefore, these nanomedicine-based phytochemicals emerge as a valuable tool to address the challenge of neurodegenerative diseases. However, it is also well known that several therapeutic approaches with competence to promote positive outcomes on animal models of neurodegenerative diseases have produced disappointing results in clinical trials [43]. Thus, we will highlight the main limitations of the present knowledge about *SC*-Nanophytosomes as a therapeutic tool for PD and the future lines of research that will be used to test their therapeutic robustness before to move it towards clinical trials.

### SC-Nanophytosomes as Therapeutic Tool for PD—Limitations and Future Line of Research

The positive outcomes of *SC*-Nanophytosomes on the rotenone-induced rat model of PD, reported here and in our previous work [16] do not ensure the therapeutic effectiveness of the formulation on human PD patients. First, rotenone is a mitochondrial toxin that promotes PD-like symptoms without significant degeneration of nigrostriatal dopamine-dependent neuronal networks, considered by many researchers as the main pathological hallmark of PD brains. Second, there remains a large gap in the link between pathological biomarkers detected in the brains of people with Parkinsonism and the signs and symptoms of the disease. Thus, *SC*-Nanophytosomes, designed to target the inner membrane of mitochondria, can mitigate mitochondrial dysfunction in brain and peripheral tissues like skeletal muscle, but this positive outcome is unable to stop or reverse the progression of the disease. Third, the available data do not allow the prediction of the bioavailability of *SC*-Nanophytosomes to the human brain and peripheral tissues, as a function of oral dose, nor their safety and tolerability profile under long-time use.

To overcome some of the above-mentioned gaps in knowledge and test the therapeutic robustness of *SC*-Nanophytosomes as a mitochondria-targeted therapy for PD, a new two-step research plan was designed to assess (i) the pharmacokinetic parameters in mouse and rat PD models by determining the *SC*-Nanophytosomes (and/or elderberry anthocyanins) levels in blood, brain, and skeletal muscle tissues, considering short- and long-term treatments, and (ii) the therapeutic efficacy of two selected doses of *SC*-Nanophytosomes on MitoPark mice, an animal model that exhibits mitochondrial dysfunction induced by genetic changes and recapitulates several other features of PD in humans, including degeneration of nigrostriatal dopamine circuitry and motor deficits [44], considering in vivo effects on motor disabilities and ex vivo endpoint analyses after 3 and 12 weeks of treatment. With this research plan, we pursue a detailed pre-clinical overview of the *SC*-Nanophytosomes to rationalize their trustworthy potential to be moved towards clinical trials.

## Figures and Tables

**Figure 1 ijms-24-16787-f001:**
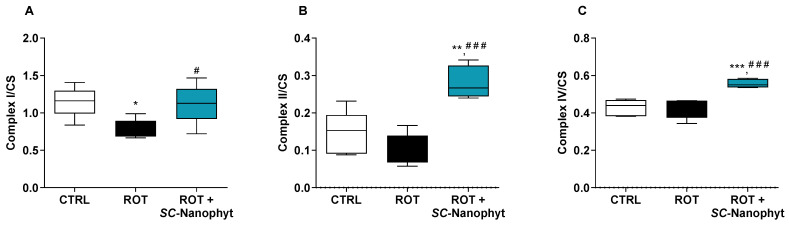
Activity of the mitochondrial respiratory complexes I (**A**), II (**B**) and IV (**C**), normalized by CS activity, in skeletal muscle mitochondria of CTRL, ROT and ROT+*SC*-Nanophyt groups. Error bars represent SEM for n = 5 independent experiments using five animals in each one. Significance was considered when *p* < 0.05. Versus CTRL: * *p* ≤ 0.05, ** *p* < 0.01 and *** *p* < 0.001. Versus ROT: # *p* ≤ 0.05 and ### *p* < 0.001.

**Figure 2 ijms-24-16787-f002:**
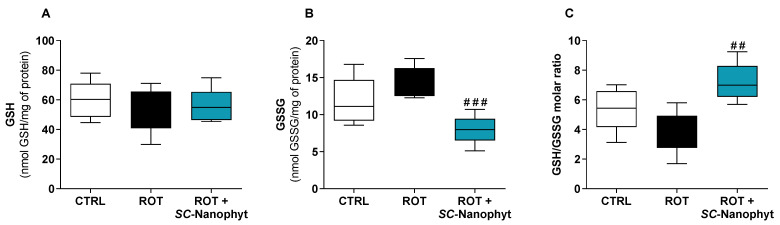
Cytosolic redox state of skeletal muscle tissues, measured by the levels of GSH (**A**) and GSSG (**B**) and GSH/GSSG molar ratio (**C**) of samples of CTRL, ROT and ROT+*SC*-Nanophyt groups. Error bars represent SEM for n = 5 independent experiments using five animals in each one. Significance was considered when *p* < 0.05. Versus ROT: ## *p* ≤ 0.01 and ### *p* < 0.001.

**Figure 3 ijms-24-16787-f003:**
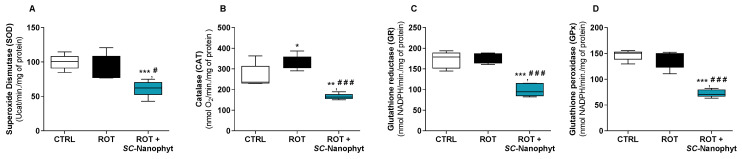
Activity of the main antioxidant enzymes, SOD (**A**), CAT (**B**), GR (**C**) and GPx (**D**), in the cytosol of skeletal muscle tissues of CTRL, ROT and ROT+*SC*-Nanophyt groups. Error bars represent SEM for n = 5 independent experiments using five animals in each one. Significance was considered when *p* < 0.05. Versus CTRL: * *p* ≤ 0.05, ** *p* < 0.01 and *** *p* < 0.001. Versus ROT: # *p* ≤ 0.05 and ### *p* < 0.001.

**Figure 4 ijms-24-16787-f004:**
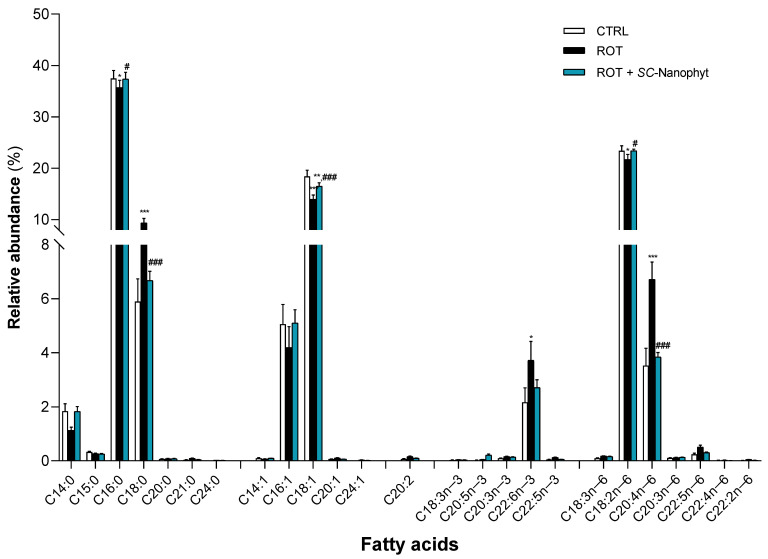
Fatty acid profile of phospholipid extracts obtained from skeletal muscle homogenates of CTRL, ROT and ROT+*SC-*Nanophyt groups (CTRL: white bars; ROT: black bars; ROT+*SC-*Nanophyt: blue bars): C14:0 = myristic acid; C15:0 = pentadecanoic acid; C16:0 = palmitic acid; C18:0 = stearic acid; C20:0 = arachidonic acid; C21:0 = heneicosanoic acid; C24:0 = lignoceric acid; C14:1 = myristoleic acid; C16:1 = palmitoleic acid; C18:1 = oleic acid; C20:1 = eicosenoic acid; C24:1 = nervonic acid; C20:2 = eicosadienoic acid; C18:3n−3 = α-linolenic acid; C20:5n−3 = eicosapentaenoic acid (EPA); C20:3n−3 = eicosatrienoate acid; C22:6n−3 = docosahexanoic acid (DHA); C22:5n−3 = docosapentaenoic acid (DPA); C18:3n−6 = γ -linolenic acid; C18:2n−6 = linoleic acid; C20:4n−6 = arachidonic acid (AA); C20:3n−6 = dihomo-γ-linolenic acid (DGLA); C22:5n−6 = docosapentaenoic acid; C22:4n−6 = docosatetraenoate; C22:2n−6 = docosadienoic acid. Error bars represent SEM for n = 5 independent experiments using five animals in each one. *, **, *** Significantly different from the CTRL group, with *p* ≤ 0.05, *p* < 0.01 and *p* < 0.001, respectively. #, ### Significantly different from the ROT group, with *p* ≤ 0.05 and *p* < 0.001, respectively.

**Figure 5 ijms-24-16787-f005:**
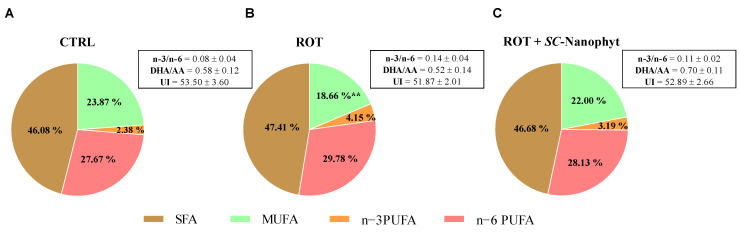
Functional parameters of the fatty acid profile obtained from total lipids obtained from skeletal muscle homogenates of CTRL (**A**), ROT (**B**) and ROT+*SC-*Nanophyt (**C**) groups. AA—arachidonic acid; DHA—docosahexaenoic acid; SFA—saturated fatty acids; MUFA—monounsaturated fatty acids; PUFA—polyunsaturated fatty acids; UI—unsaturation index. ** Significantly different from the CTRL group, with *p* < 0.01.

## Data Availability

Data are contained within the article and Appendix A.

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
