# Peer review of "Mitochondrial Dysfunction in Skeletal Muscle of Rotenone-Induced Rat Model of Parkinson’s Disease: SC-Nanophytosomes as Therapeutic Approach"

_ijms, 2023, doi:10.3390/ijms242316787_

Round 1

Reviewer 1 Report

Comments and Suggestions for Authors

I am grateful for the opportunity to review this paper. The authors have conducted a study in a rat model with rotenone-induced PD, and separated into 3 groups. The main objective is to address the mitochondria as a therapeutic target in PD.

In the Introduction the authors provide adequate and updated information. The references used are coherent and favour the reader's understanding of the object of study.

It has been studied in vivo and ex vivo. The procedures are adequate and consistent with the results obtained.

In relation to the figures, I consider that they should be published in colour or at least modify the current tones, as information is lost as they are very similar (for example, figure 4 in lines 274-275).

Figure 5 should be modified and made clearer and of higher quality for publication.

It is recommended to include a section on Limitations of the study and another section on Future lines of research. In an experimental context, it is important to leave these reflections for researchers consulting the manuscript.

Author Response

Answer to reviewers

We are grateful for reviewers’ careful manuscript appreciation, and we will try our best to address all the suggestions proposed. The modifications are highlighted in track changes in the new version of the manuscript.

Reviewer #1

General comment: “I am grateful for the opportunity to review this paper. The authors have conducted a study in a rat model with rotenone-induced PD and separated into 3 groups. The main objective is to address the mitochondria as a therapeutic target in PD. In the Introduction the authors provide adequate and updated information. The references used are coherent and favour the reader's understanding of the object of study. It has been studied in vivo and ex vivo. The procedures are adequate and consistent with the results obtained.”

Comment 1 – “In relation to the figures, I consider that they should be published in colour or at least modify the current tones, as information is lost as they are very similar (for example, figure 4 in lines 274-275). Figure 5 should be modified and made clearer and of higher quality for publication.”

Answer to Comment 1:

As suggested, the quality of the figures was improved, and the current grey scale was changed for different colors.

Comment 2: “It is recommended to include a section on Limitations of the study and another section on Future lines of research. In an experimental context, it is important to leave these reflections for researchers consulting the manuscript.”

Answer to Comment 2:

We appreciate the relevance of the reviewer's comment, that is recognized as an opportunity to extend the scientific relevance of the manuscript. It is well-known that several therapeutic approaches with competence to promote positive outcomes on animal models have produced disappointing results in clinical trials. Thus, a scientific rationalization for a putative failure in clinical trials of any successful therapeutic strategy in an animal model can (and should) be subject of reflection before to be moved for clinical trials. Accordingly, the conclusion was rearranged and the main limitations of the present knowledge about SC-Nanophytosomes as therapeutic tool for PD as well as the future lines of research were included in a new subtopic of the conclusions, as follow:

“…

Therefore, this nanomedicine-based phytochemicals emerge as valuable tool to address the challenge of neurodegenerative diseases. However, it is also well-known that several therapeutic approaches with competence to promote positive outcomes on animal models of neurodegenerative diseases have produced disappointing results in clinical trials [43]. Thus, we will highlight the main limitations of the present knowledge about SC-Nanophytosomes as therapeutic tool for PD and the future lines of research that will be used to test their therapeutic robustness before to move it towards clinical trials.

4.1. SC-Nanophytosomes as therapeutic tool for PD – Limitations and future line of research

The positive outcomes of SC-Nanophytosomes on rotenone-induced rat model of PD, reported here and in our previous work [16], do not ensure the therapeutic effectiveness of the formulation on human PD patients.  First, rotenone is a mitochondrial toxin that promotes PD-like symptoms without significant degeneration of nigrostriatal dopamine-dependent neuronal networks, considered by many researchers as the main pathological hallmark of PD brains. Second, there remains a large gap in the link between pathological biomarkers detected in the brains of people with Parkinsonism and the signs and symptoms of the disease. Thus, SC-Nanophytosomes, designed to target the inner membrane of mitochondria, can mitigate mitochondrial dysfunction in brain and peripheral tissues like skeletal muscle, but this positive outcome is unable to stop or reverse the progression of the disease. Third, the available data do not allow to predict the bioavailability of SC-Nanophytosomes to the human brain and peripheral tissues, as function of oral dose, nor their safety and tolerability profile under long-time use.

To overcome some of the above-mentioned gap of knowledge and test the therapeutic robustness of SC-Nanophytosomes as mitochondria-targeted therapy for PD, a new two-step research plan was designed to assess: i) the pharmacokinetic parameters in mouse and rat PD models by determining the SC-Nanophytosomes (and/or elderberry anthocyanins) levels in blood, brain, and skeletal muscle tissues, considering short- and long-term treatments, and ii) the therapeutic efficacy of two selected doses of SC-Nanophytosomes on MitoPark mice, an animal model that exhibit mitochondrial dysfunction induced by genetic changes and recapitulate several other features of PD in humans, including degeneration of nigrostriatal dopamine circuitry and motor deficits [44], considering in vivo effects on motor disabilities and ex vivo endpoint analyses after 3 and 12 weeks of treatment. With this research plan we pursue a detailed pre-clinical overview of the SC-Nanophytosomes to rationalize about their trustworthy potential to be moved towards clinical trials.”

Reviewer 2 Report

Comments and Suggestions for Authors

The manuscript 'Mitochondrial dysfunction in skeletal muscle of rotenone-induced rat model of Parkinson’s disease: SC-Nanophytosomes as therapeutic approach' is written well and the methodologies outlined are appropriate. 

Please find my comments.

1. Line 94-97 - misfolded synuclein in skeletal muscles of PD patients?

2.  Figure 1,2,3,4,5 - graphs can be improved.

3. Line 373-375 - You performed spectrofluorimetric assay to evaluate the mitochondrial redox chain complex. How this assay is different from the Seahorse assay?

Comments on the Quality of English Language

Minor typo errors.

Author Response

Answer to reviewers

We are grateful for reviewers’ careful manuscript appreciation, and we will try our best to address all the suggestions proposed. The modifications are highlighted in track changes in the new version of the manuscript.

Reviewer #2

General comment: The manuscript 'Mitochondrial dysfunction in skeletal muscle of rotenone-induced rat model of Parkinson’s disease: SC-Nanophytosomes as therapeutic approach' is written well, and the methodologies outlined are appropriate.

Please find my comments.

Comment 1. Line 94-97 - misfolded synuclein in skeletal muscles of PD patients?

Answer to Comment 1:

We appreciate the reviewer's question since it allows to correct an inaccuracy in the manuscript's introduction. In fact, the misfolded alpha-synuclein was not detected in skeletal muscles tissues of PD patients, but in patients of Sporadic Inclusion-body Myositis - a common progressive muscle disease of older patients, as indicated by the reference 20 in the first version of the manuscript. Thus, this reference was eliminated, and the corresponding sentence was shortened, as follow:

“Although the PD symptoms are mainly associated with the degenerative brain process, the skeletal muscle of PD patients also exhibited several brain pathological hallmarks of the disease, including mitochondrial dysfunction related to deficient activity of the mitochondrial redox chain [18,19].”

Comment 2Figure 1,2,3,4,5 - graphs can be improved.

Answer to Comment 2:

As suggested, the quality of the figures was improved, and the current grey scale was changed for different colors.

Comment 3. Line 373-375 - You performed spectrofluorimetric assay to evaluate the mitochondrial redox chain complex. How this assay is different from the Seahorse assay?

Answer to Comment 3:

The assessment of the activities of the mitochondrial redox chain complexes provides information about the maximum activity of the individual components of mitochondrial respiratory chain. It can be assessed in mitochondria obtained from fresh or frozen tissues submitted to several cycles of freezing/thawing to allow substrate assessment to active center of the enzyme into inner mitochondrial membrane or matrix. On the other hand, the Seahorse microplate technology allows to obtain information about the mitochondrial-related metabolic fluxes (e.g., oxygen consumption rate), but it requires living cell or intact mitochondria isolated from fresh tissue samples. In fact, the mitochondrial-related metabolic fluxes depend not only the activity of the mitochondrial respiratory complexes and F0F1-ATPsynthase but also of other factors like the cellular ATP/ADP ratio. Thus, the information obtained with activities of the mitochondrial redox chain complexes and Seahorse microplate technology are complementary.

Since the number of different samples that can be processed in each endpoint day (animals sacrificed) the mitochondrial-related metabolic fluxes were assessed in brain mitochondria using Oroboros Technology (data in our previous work), while the skeletal muscles tissues were collected and frozen with liquid nitrogen and stored at – 80 ºC until to be used for the present study.

For the above, we did not consider suitable to include in the manuscript a comparison between the obtained values of the activity of the mitochondrial redox chain complexes and the information that can be obtained with Seahorse microplate technology, not used in the present work.
